# Dynamic Motions of Piled Floating Pontoons Due to Boat Wake and Their Impact on Postural Stability and Safety

Elizabeth L. Freeman, Kristen D. Splinter *, Ron J. Cox and Francois Flocard

Water Research Laboratory, School of Civil and Environmental Engineering, UNSW Sydney,
Sydney 2093, Australia
* Correspondence: k.splinter@unsw.edu.au

**Abstract:** Piled floating pontoons are public access structures that provide a link between land and sea. Despite floating pontoons being frequented by the public, there is limited data available to coastal or maritime engineers detailing the dynamic motions (acceleration and rotation) of these structures under wave action and the impact of these motions on public comfort and safety to inform their design. This contribution summarises results from a set of laboratory-scale physical model experiments of two varying beam width piled floating pontoons subjected to boat wake conditions. Observed accelerations and roll angles were dependent on beam-to-wavelength ratio (B/L), with the most adverse motion response observed for B/L ~0.5. Internal mass of the pontoon played a secondary role, with larger mass structures experiencing lower accelerations for similar B/L ratios. Importantly, these new experimental results reveal the complex interaction between the piles and pontoon that result in peak accelerations more than six times the nominated operational safe motion limit of 0.1g. Root mean square (RMS) accelerations were more than three times the nominated comfort limit (0.02g) and angles of rotation more than double what would be perceived as safe (6 degrees) for the boat wake conditions tested. The frequency of acceleration also suggests patrons standing on these platforms are likely to experience discomfort and instability. Laboratory results are compared against a series of field-scale experiments of pontoon motion response and patron feedback. The dynamic motion response of pontoons tested in both field-scale and laboratory experiments compared well.

**Keywords:** floating bodies; piled structure; safe motion limits; peak acceleration; root mean square acceleration; angle of rotation; personal stability; Inertial Measurement Units; maritime structures

## 1. Introduction

Many sheltered, small craft harbours around the world utilize floating pontoons as landing stages for vessel passengers, pedestrians, small cargo, roll-on roll-off (Ro-Ro) berths, and for mooring small boats [1,2]. In Sydney Harbour (Australia) alone, there are more than 137 public access points (wharves, jetties, and pontoons) for boat users frequented by more than 172,000 commuter passengers and tourists per month [3] (Figure 1). Floating pontoons have several advantages over fixed public access structures, including cost, ease of installation, and flexibility in dynamic water level environments [2]. However, little research has been done on floating pontoons to understand their dynamic stability with respect to patron safety. This contrasts with the significant research undertaken by coastal and ocean engineers on the hydrodynamic performance of similar structures, such as floating breakwaters, or on fixed platforms. Floating breakwaters differ from pontoons in that their primary design consideration is the dissipation and reflection of wave energy, rather than providing safe public access [4–8]. Fixed platforms, such as decks that are used for public access do not dynamically move but may be subject to large uplift forces from waves [9,10], and are typically designed based on wave loads (forces) rather than motion response. As such, limited availability of clear design guidelines for floating piled

pontoons can result in undesirable dynamic behaviour and reduced usability, incurring costly retrofitting to operators.

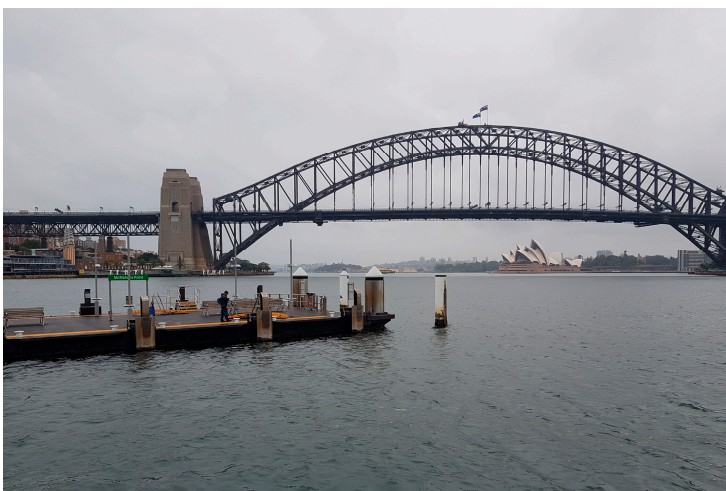

**Figure 1.** A ferry commuter floating pontoon located at McMahons Point in Sydney Harbour, Australia.

To date, there has been limited research on the dynamic motions of floating pontoons secured by piles despite these types of pontoons being actively installed and used worldwide as public access structures (Figure 1). Given the lack of research into floating pontoons, here we summarize recent research relevant to floating breakwaters. The high inspection and maintenance costs associated with flexible mooring systems have resulted in a clear preference for piled restraining systems in sheltered harbours of depths less than 10 m [6]. While several studies have documented the amplitudes (displacement) of motion for heave and surge as well as rotation angles of floating breakwaters under both regular and irregular wave action [5,11–14] they do not report on the dynamic motions (accelerations) of the structures which is a key design consideration with regard to postural stability and public safety of public access structures [15]. Many numerical studies examining the hydrodynamic problem of floating breakwaters have also been conducted [5,16–18]. However, these too have not focussed on the design problem of piled pontoon systems and/or reported on the observed accelerations, despite these being popular coastal infrastructure in sheltered wave environments used for public access.

Related to the new experimental data presented in this paper, Cox et al. [6] conducted a series of scaled physical laboratory experiments to examine the effect of both monochromatic and irregular waves on the dynamic motions of a piled floating pontoon breakwater. They reported peak vertical accelerations ranging from 0.1g up to approximately 2.25g for a prototype wave height of 0.4 m. Although they did not report accelerations for the larger wave heights tested it was observed that both the vertical and roll motions of the piled floating breakwater were of greater magnitude and more violent when subjected to larger waves.

This paper presents laboratory-scale experimental results on the dynamic motions of piled floating pontoons under boat wake conditions and compares them to a set of Safe Motion Limit (SML) criteria determined to ensure postural stability and comfort of patrons. In Section 2, the concepts of postural stability with respect to accelerations and angles of motion are summarised. The SML criteria are then established to ensure postural stability is maintained for able-bodied adults standing on floating structures. Details of the new laboratory experiments are provided in Section 3. Section 4 presents the results of the measured dynamic motions, followed by discussion of the potential impact these dynamic motions could have on a person's postural stability and a comparison to preliminary field data in Section 5.

## 2. Postural Stability and Comfort

Postural stability is a person's ability to maintain the body's centre of gravity over a base support during quiet standing and movement [19]. However, postural stability is a complex, biomechanical process that involves coordinated actions of the sensory, motor, and central nervous system [20] and it varies with the age of the subject [19,21–23]. Young children (<7 years) and the elderly (>65 years) have lower stability limits than those between the ages of 7–65 years [19,21,23,24]. With respect to floating pontoons as public spaces, postural stability is discussed in reference to vertical and lateral accelerations, angular rotation, and the frequencies at which these occur. For a more detailed review on postural stability, the reader is referred to [15]. As piled floating pontoons are commonly used by the public, postural stability and safety should be considered at the design stage. Here, we summarize relevant research and standards related to the dynamic motions of marine structures and human stability.

### 2.1. Root Mean Square and Peak Acceleration

Based on the general effects of motion on human performance, [25] presented RMS acceleration limits for transit passengers (0.05g) and cruise liners (0.02g). Specific to cruise liners, NORDFORSK [26] separated out the RMS criteria into lateral (0.02g) and vertical (0.03g) limits for passengers to remain comfortable. STANAG 4154 [27] specified a safety and effectiveness limiting criteria of peak vertical and lateral accelerations of 0.2g and 0.1g, relative to the bridge of the naval vessel. For a floating pontoon relative to serviceability, ref. [28] suggested a peak acceleration of 0.1g. de Graaf et al. [29] used a moving treadmill to investigate the limits of acceleration the human body can withstand without losing balance. Their study found that participants were most vulnerable to sideways acceleration and least vulnerable to backwards acceleration. They found that a standing person could endure a maximum forward acceleration of 0.054g, maximum sideward acceleration of 0.045g and a maximum backward acceleration of 0.061g before losing balance. These values are much lower than those proposed by [27,28] with respect to the serviceability of structures.

### 2.2. Frequency of Acceleration

Humans are more likely to have unfavourable response to motion within a frequency band of 1–80 Hz [30–32]. Nawayseh et al. [31] found loss of balance increased with increasing magnitude of horizontal and rotational oscillation. They found that loss of balance and subjective estimates of the probability of losing balance all peaked at around 0.5 Hz, while fore-and-aft oscillation caused more instability than lateral oscillation. They also found that standing people were more sensitive to low-frequency (0.125–2.0 Hz) acceleration when exposed to translational oscillation but were more sensitive to high frequency (2 Hz) acceleration when exposed to the gravitational acceleration arising from rotation (i.e., pitch or roll).

Motion sickness is another design consideration that occurs for motions between 0.1–0.5 Hz [33,34]. For improving passenger comfort and to reduce the incidence of motion sickness to 10% or less, [34] recommend a maximum RMS acceleration value of 0.007g. Shupak and Gordon [35] identified the greatest incidence of seasickness was found at a frequency of 0.2 Hz, increasing with the acceleration level from a threshold value of 0.01g.

### 2.3. Angles of Motion

As well as the magnitude and frequency of acceleration, the angles of motion of a dynamic body need to be considered for the comfort and safety of the users. For small craft harbours, [36] nominated a maximum roll angle (*longitudinal*-axis) of 6° and a maximum angular acceleration of 2°/s² for floating breakwaters or pontoons. Stevens and Parsons [25] reported a maximum allowable RMS roll of 2.5° and 2.0° for transit passengers and cruise liners, respectively, while [26] limits RMS roll to 2.0°.

### 2.4. Safe Motion Limit Criteria Adopted for this Study

For this study, the Safe Motion Limits (SML) related to postural stability of a person with respect to dynamic motions of a piled floating pontoon are defined in Table 1. Based on the literature summarized above, dynamic motions exceeding these limits have the potential to result in motion sickness, postural instability, safety, fatigue, and discomfort.

**Table 1.** Safe Motion Limits (SML) adopted for this study as relevant for older children and adults (ages 7–65 years).

| Criteria | Limit | Reference |
|:---:|:---:|:---:|
| **Operation (Peak values)** | | |
| Peak Vertical Acceleration | 0.1g | [28] |
| Peak Lateral Acceleration | 0.1g | [27,28,37] |
| Peak Angle of Tilt | 6 degrees | [36] |
| **Comfort (RMS values)** | | |
| RMS Vertical Acceleration | 0.02g | [25,26] |
| RMS Lateral Acceleration | 0.03g | [26] |
| RMS Roll | 2° | [25,26] |

## 3. Methods

### 3.1. Experimental Setup

The physical model testing was conducted in the 0.6 m wide wave flume at the Water Research Laboratory at UNSW Sydney. The flume dimensions were 30 m long, 0.6 m wide and 0.7 m deep. Froude similitude is commonly applied in hydraulic structure model scale testing and was applied between prototype and model conditions using a length scale of 10 for these experiments in agreement with previous studies [6,7]. Froude scaling is the most appropriate for these set of experiments for several reasons, including that it emphasizes the inertial and gravity forces, with the tests here focussing on the dynamic motions (accelerations) of the structure. In Froude similitude, accelerations have a scale factor = 1. The models tested were two piled rectangular floating pontoons of varying prototype beam widths subsequently referred to as Narrow (2.83 m beam) and Wide (5.63 m beam). A narrow gap between the side wall of the flume and pontoon ensured that no collisions existed between the flume wall and pontoon [38] and limited any side wall effects [39].

The floating pontoons have six degrees of freedom, here defined as follows: surge (in the direction of wave propagation, $x_b$), sway (perpendicular to the direction of wave propagation, $y_b$) and heave (vertical, $z_b$), as well as the three rotations around the centre of gravity (roll ($\phi$), pitch($\theta$) and yaw ($\psi$)). Here, roll is taken as the rotation around the longitude (longest) axis (y), pitch is around the shorter pontoon axis (x), and yaw is around the vertical axis (z). Several physical characteristics of the pontoons influence the stability and dynamic motion of floating bodies, including the draft-to-water-depth ratio (D/d), the structure-beam-to-draft ratio (B/D), and the beam-to-wavelength ratio (B/L), the metacentric height (*GM*), the radius of gyration (*K*), as well as the wave direction and the degree of mooring restraint [40]. These are summarized in Table 2. The metacentric height is the vertical distance between the centre of gravity (c.g.) and the Metacentre (*M*) and is calculated as follows:

$$GM = KB + BM - KG \tag{1}$$

where *KB* is the vertical distance from keel to centre of buoyancy (c.b.) in metres and is equal to the exact middle of the volume of displaced water. *BM* is the vertical distance from the centre of buoyancy (c.b.) to the metacentre (*M*), and *KG* is the vertical distance from the keel to the centre of gravity (c.g.). According to [40], the radius of gyration (*K*) for a floating pontoon is between 0.29 B and 0.35 B, where B is the beam. Here, *K* in the roll

direction is theoretically derived from the inertia of the water plane area (*I*) and the area of contact (*A*), where:

$$K = \sqrt{I/A} = {}^{B}/\sqrt{12} = 0.29B \qquad (2)$$

and represents the lower bound proposed by [40]. As the pontoon is attached by piles on the seaward side, pitch and yaw movements are highly restrained and not included here.

**Table 2.** Physical characteristics of the two pontoons tested. All values provided in prototype.

| Pontoon Measurements | Narrow Pontoon | Wide Pontoon |
|---|---|---|
| Beam [m] | 2.83 | 5.63 |
| Length [m] | 5.59 | 5.59 |
| Draft [m] | 0.455 | 0.455 |
| Metacentric Height (GM) [m] | 1.23 m | 5.58 m |
| Radius of Gyration, roll (K) [m] | 0.83 | 1.63 |
| Displacement [Tonnes of water displaced] | 7.41 tonnes | 14.75 tonnes |

The pontoon models were constructed of grey PVC sheet, with additional PVC sheet used for internal ballast to alter the draft of the pontoon [41]. The pontoons were connected to two, 330 mm diameter (prototype) vertical piles located on the seaward side (Figure 2). Delrin, a highly crystalline engineering thermoplastic specified for high load mechanical applications, was used to construct wear/impact buffers at the pontoon/pile interface. These buffers provided a low friction sliding connection between the restraining piles and the pontoon. The pontoon/pile connections (42 mm clearance at prototype scale between pile and collar) allowed free vertical movement and restrained (but measurable) lateral movement in the absence of waves.

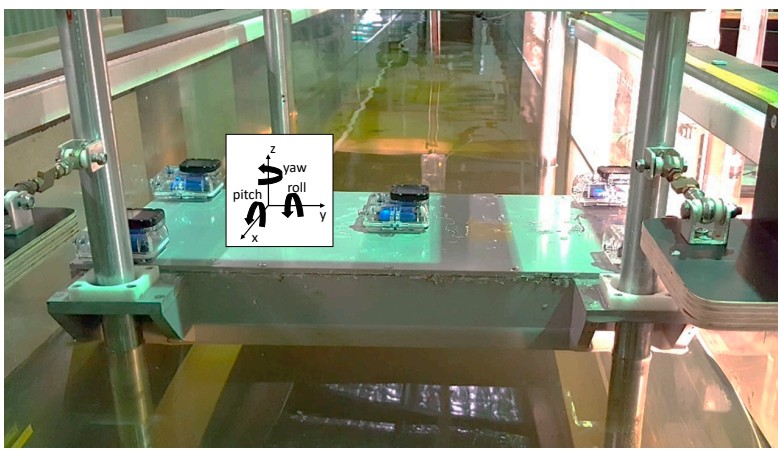

**Figure 2.** Photo of Narrow Pontoon Showing Seaward Face, Pile Mooring System and Positioning of Five IMU. The 6-degrees of freedom are also shown.

On each pontoon, five Life Performance Research (LPMS-B2) Inertial Measurement Units (IMU) were used to measure triple-axis accelerations and triple-axis angles of each floating pontoon [42]. The IMUs were contained in GoPro housing for waterproofing with double sided tape inside to secure them in place. Each GoPro case was secured to the pontoons using adhesive Velcro located on each corner (Sensors 1–4, Figure 2) as well as the centre top face (Sensor 5, Figure 2).

According to the technical specifications provided by LP-Research, each IMU uses a 3-axis gyroscope (used to measure angular velocity), a 3-axis accelerometer (used to detect the direction of the earth's gravitational field) and a 3-axis magnetometer (to measure the direction of the earth's magnetic field). The orientation data recorded by the gyroscope

is corrected with information from the accelerometer (roll and pitch directions) and the magnetometer (yaw direction). The accelerations recorded were in units of g (gravity, 9.81 m/s$^2$). The units were able to measure orientation in 360 degrees about all three axes, where z is in the direction of earth's gravity (vertically down with $-1$ g), x- in the direction of wave propagation and y- in the cross-tank direction, following a right-handed Cartesian coordinate system.

Unless otherwise stated, all default settings of the IMUs were employed. Calibration of each IMU requires determination of the gyroscope bias, gain, and movement threshold, as well as accelerometer misalignment, offset, gain, and magnetometer interference bias and gain. As the gyroscope sensor has a constant bias that may be influenced by environmental factors, such as temperature, gyroscope bias calibration was undertaken for each round of testing using manual calibration whereby the sensors were placed in a motionless state and firmware command used to trigger gyroscope calibration.

The internal sampling and filtering of the IMU is 400 Hz. Bluetooth connection between the IMUs and the log computer was used to allow for immediate data recording of accelerations and rotations of the floating pontoons as the motions took place. Data was recorded at a rate of 50 Hz. Data recording at a rate above this caused Bluetooth connection errors. This rate was considered sufficient to capture the relevant frequencies as described above. All IMUs were synchronised to record at the same time.

From the IMUs output files, the following indices were used for the reported results: Surge Accelerations: LinAccX(g); Sway Accelerations: LinAccY(g); Heave Accelerations: LinAccZ(g); and Roll Angles: EulerY(degrees). At the beginning of each experiment after the IMUs were attached to the pontoon, turned on and synced, the initial ~20 s of the output file was used to confirm each sensor was working properly and the mean of the accelerometers was adjusted so that it was (x,y,z) = (0,0,0) during post-processing.

### 3.2. Natural Periods of Motion

The response of floating bodies to waves is highly dependent on wave period and wavelength, with the maximum response likely to occur when the wave period coincides with the natural frequency of motion of the structure or when the wavelength coincides with twice the structure's length [2]. Decay tests of the piled pontoons were carried out in still water and motions recorded using the five IMU positioned centrally and on the corners of the pontoons (Figure 2). The decay tests were undertaken in heave by pushing the pontoons down so there was no freeboard and releasing. This was done three times for each pontoon and results analysed to determine the natural period. The same was done for roll by inclining the pontoons approximately 20° and releasing while the IMU recorded the motions. The time between adjacent crests/troughs was determined for the natural roll period.

### 3.3. Wave Environment

Waves were generated by a piston-type wave paddle situated at one end of the flume. A mild-slope (1 V:10 H) dissipative beach fitted with high-density reticulated foam was installed at the other end of the wave flume to minimise wave reflection. Three wave probes were set up between the wave paddle and the tested pontoon. A fourth probe was positioned in the lee of the pontoon to measure the transmitted wave. The wave heights and periods used for the test program were representative of boat wake conditions in Sydney Harbour [43], where there are over 100 public access points such as wharves, jetties, and pontoons [3]. Test conditions are summarized in Table 3. All dimensions and times given are prototype values unless otherwise specified. Triplicate runs, of a duration of 189 s, were conducted for each of the wave periods to ensure similarity between tests. All tests were completed in a water depth (d) of 3.6 m and both pontoons had a draft (D) of 0.45 m, (D/d = 0.125). Standard three probe array reflection analysis [44] did not provide robust results due to the short record length imposed by the time interval taken by reflected waves

to reach the wave paddle. Therefore, a multi-step process of signal analysis was developed to determine the incident and reflected wave heights from the recorded timeseries.

**Table 3.** Monochromatic Wave Testing Parameters (Prototype Scale).

| Test ID | Wave Period T (s) | Wave Height H (mm) | Beam B(m) | Draft D(m) | Depth d(m) |
|---------|-------------------|--------------------|-----------|------------|------------|
| B1 | 2 | 300 | 2.83 | 0.45 | 3.6 |
| B2 | 3 | 310 | 2.83 | 0.45 | 3.6 |
| B3 | 5 | 290 | 2.83 | 0.45 | 3.6 |
| B4 | 7 | 320 | 2.83 | 0.45 | 3.6 |
| B5 | 2 | 300 | 5.63 | 0.45 | 3.6 |
| B6 | 3 | 310 | 5.63 | 0.45 | 3.6 |
| B7 | 5 | 290 | 5.63 | 0.45 | 3.6 |
| B8 | 7 | 320 | 5.63 | 0.45 | 3.6 |

First, a Savitzky-Golay filter third-order polynomial was applied to eliminate any high frequency noise. Then, the incident wave timeseries was determined from the first portion of the recorded timeseries prior to wave reflections off the pontoon structure at each of the three probes. This comprised approximately 4–14 waves depending on the wave period being analysed. Wave height and period were first estimated using the zero-crossing method. An exhaustive search was done around the estimated wave to generate the optimum incident wave signal based on cross-correlation analysis of the measured and generated free surface in the time domain. This was done to determine the best fit wave period. The reflected wave free surface ($\eta_r$) was then determined using the relationship $\eta = \eta_i + \eta_r$ from the latter portion of the raw wave probe time series comprising approximately 5–19 waves depending on the wave period. This was completed for each of the three trials for each wave period and each probe (i.e., single probe analysis). A representative time slice of the incident waves for each of the four wave periods tested is provided in Figure 3.

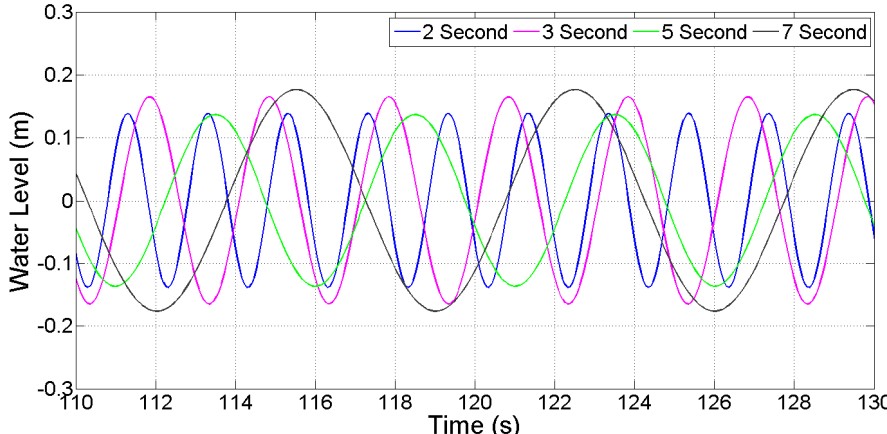

**Figure 3.** Example incident band water surface elevation timeseries, $\eta_i$, for each of the four wave periods tested. Prototype scale.

## 4. Results

Results are presented for both the Narrow and Wide Pontoons relative to beam-to-wavelength ratios determined from Table 3. The natural periods of heave and roll were calculated as the mean of the three plunge tests and summarized in Table 4. The natural periods are between 2 and 3 s (prototype) and do not directly coincide with any of the tested incident wave periods and therefore should not cause adverse dynamic motions.

**Table 4.** Summary of Experimental Natural Period (prototype) in Heave and Roll for Both the Narrow and Wide Pontoons Constrained by Piles.

| Natural Period (s) | Narrow | Wide |
|---|---|---|
| $T_N$—heave | 2.44 | 2.61 |
| $T_N$—roll | 2.91 | 2.64 |

### 4.1. Operational Criteria: Reflection and Transmission Coefficients

As noted in Section 3.1, dynamic motions of floating bodies are highly dependent on the relative draft-to-depth ratio (D/d), the structure beam-to-draft ratio (B/D), and the beam-to-wavelength ratio (B/L), as well as the wave direction and the degree of mooring restraint [40]. Transmission ($K_t = \frac{H_t}{H_i}$) and reflection ($K_r = \frac{H_r}{H_i}$) coefficients, where $H_t$ is the transmitted wave height, $H_i$ is the incident wave height and $H_r$ is the reflected wave height are summarized in Table 5 and Figure 4. Figure 4a shows wave transmission relative to wave period and includes comparative results from similar studies [6,7]. Their tests were of similar beam width (2.4 m and 4.8 m), however had much larger draft (1.7 m compared with 0.45 m used in the present study) and larger waves (0.2–1.2 m), with similar wave periods (2–5 s) and water depth (4.2 m).

**Table 5.** Monochromatic Wave Testing Results Including Transmission and Reflection Coefficients (Prototype Scale).

| Test ID | Wavelength L (m) | Wave Steepness H/L | Water Depth to Wavelength d/L | Beam-to-Wavelength B/L | Transmission $K_t = H_t/H_i$ | Reflection $K_r = H_r/H_i$ |
|---|---|---|---|---|---|---|
| B1 | 6.23 | 0.048 | 0.58 | 0.45 | 0.38 | 0.57 |
| B2 | 13.17 | 0.024 | 0.27 | 0.22 | 0.94 | 0.10 |
| B3 | 26.85 | 0.011 | 0.13 | 0.11 | 0.93 | 0.15 |
| B4 | 39.59 | 0.008 | 0.09 | 0.07 | 0.94 | 0.20 |
| B5 | 6.23 | 0.048 | 0.58 | 0.90 | 0.26 | 0.58 |
| B6 | 13.17 | 0.024 | 0.27 | 0.43 | 0.96 | 0.13 |
| B7 | 26.85 | 0.011 | 0.13 | 0.21 | 0.93 | 0.17 |
| B8 | 39.59 | 0.008 | 0.09 | 0.14 | 0.99 | 0.21 |

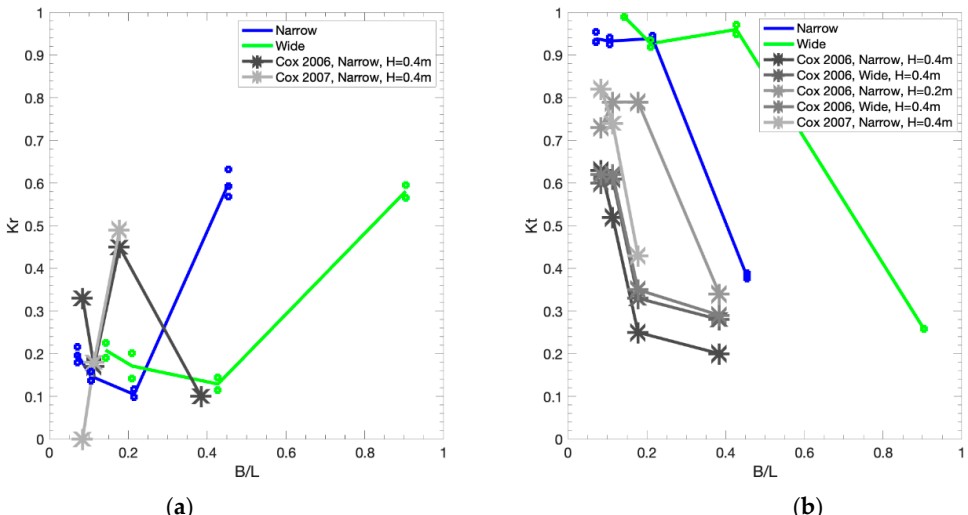

(**a**)                    (**b**)

**Figure 4.** Variation of (**a**) Reflection and (**b**) Transmission Coefficients versus Beam-to-wavelength (B/L) for the Narrow and Wide Pontoon based on monochromatic boat wake (H = 0.3 m) for a draft of 0.455 m. Repeat tests are included as dots with lines representing the average. For comparison, available Kt and Kr values for similar studies [6,7] are included, noting that the draft in [6,7] was 1.7 m and this affects the movement of the structure.

While the results are presented as a function of B/L in Figure 4, there is a clear dependency on wave period indicated by the lateral shift in the Narrow and Wide results. The highest reflection ($K_r$ = 0.60 and 0.58) occurred during the 2 s period wave (Figure 4a) with both pontoons experiencing strong interaction with the incoming waves and the fixed piles that resulted in shock accelerations as they were pushed against the pile. For longer wave periods (lower B/L values) the pontoons rode over the waves and reflection coefficients were <0.2. As shown in Figure 4b, at 2 s both pontoons tested observed effective attenuation performance with slightly better attenuation for the wider pontoon. Transmission was strongly dependent on wave period with performance being significantly reduced for wave periods greater than 2 s. For wave periods of 3 s or above, beam had minimal effect on $K_t$ for the new tests presented here compared to previous work. Larger drafts as used in the Cox 2006 and 2007 papers resulted in more effective attenuation (smaller Kt values) of waves for similar B/L.

### 4.2. Operational Criteria: Peak Vertical and Lateral Acceleration

Considering patron safety and stability of floating pontoons, peak accelerations are a key operational criterion as large and short duration spikes are likely to cause loss of balance. As anticipated, the dynamic motions of the piled pontoons varied with wave period and pontoon width. There was significant wave-structure interaction (and high energy losses) that produced higher accelerations frequently exceeding the operational SML of 0.1g for the Narrow Pontoon (Figure 5a,b) compared to the Wide Pontoon (Figure 5c,d). In agreement with the Kr and Kt values presented in Figure 4, for longer wave periods (lower B/L), both pontoons acted slightly more like a floating vessel, riding over the waves, experiencing less wave-structure interaction and smaller spikes in both vertical and lateral acceleration (Figure 5).

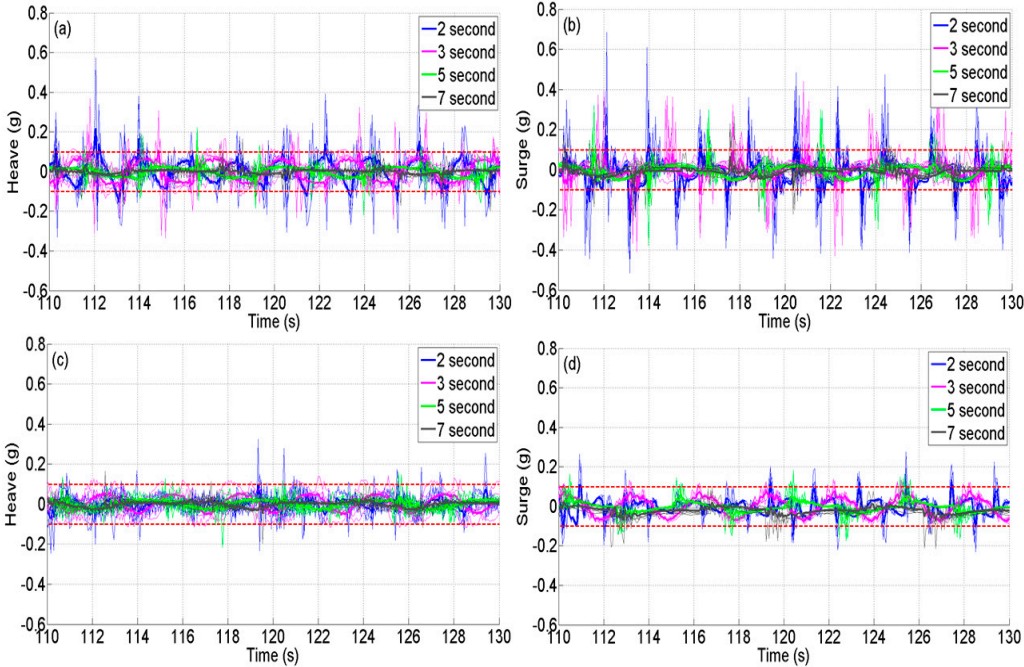

**Figure 5.** Twenty-second time slice of raw acceleration versus time: (**a**) Narrow Pontoon Heave Acceleration; (**b**) Narrow Pontoon Surge Acceleration; (**c**) Wide Pontoon Heave Acceleration; and (**d**) Wide Pontoon Surge Acceleration. The horizontal red dashed line indicates the Safe Motion Limit of 0.1 g.

Short duration spikes in heave and surge acceleration occurred at both the crest and trough of the wave for shorter waves, particularly when combined with the lower B/D ratio of the Narrow pontoon (Figure 5a,b vs. Figure 5c,d). To further understand the cause of these spikes, a 5 s time slice is presented in Figure 6. At the crest of the wave, the pontoon

is visibly pushed against the piles creating impact acceleration spikes in surge (Figure 6a,e). In some instances, the pontoon is observed to hang briefly on the pile due to the high roll angles (Figure 6d), leading to both high heave and surge accelerations when the pontoon subsequently falls and impacts the piles at the base of the wave (Figure 6c,e). Comparison of these observations to field cases are discussed in Section 5.1.

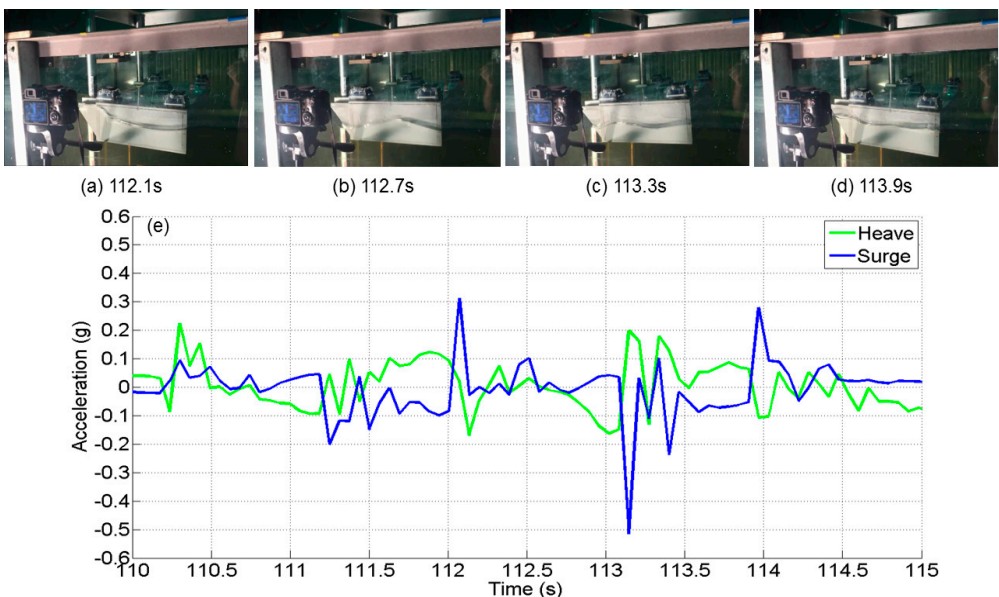

**Figure 6.** Time slice of Narrow Pontoon during the 2 s wave test. (**a–d**) Snap shots of pontoon motion and Sensor 1 heave and surge accelerations.

Considering the full time period of each experimental run (approximately 189 s), peak heave (z-axis) and surge (x-axis) accelerations (0.58g and 0.65g, respectively), were as high as six times the peak SML (0.1g) while sway (y-axis) peak accelerations reached three times the SML (0.32g) for the Narrow pontoon (Figure 7). All peak accelerations exceeded the safe motion limits, with the highest accelerations recorded for the 2 s period wave (B/L ~ 0.45, Narrow, Figure 7a and B/L ~ 0.90, Wide, Figure 7b). For similar B/L, lower B/D ratios resulted in higher peak accelerations. Additionally, peak accelerations showed a stronger dependence on B/L for the Narrow Pontoon compared to the Wide Pontoon (Figure 7a vs. Figure 7b). The results presented here are in agreement with previous studies [6].

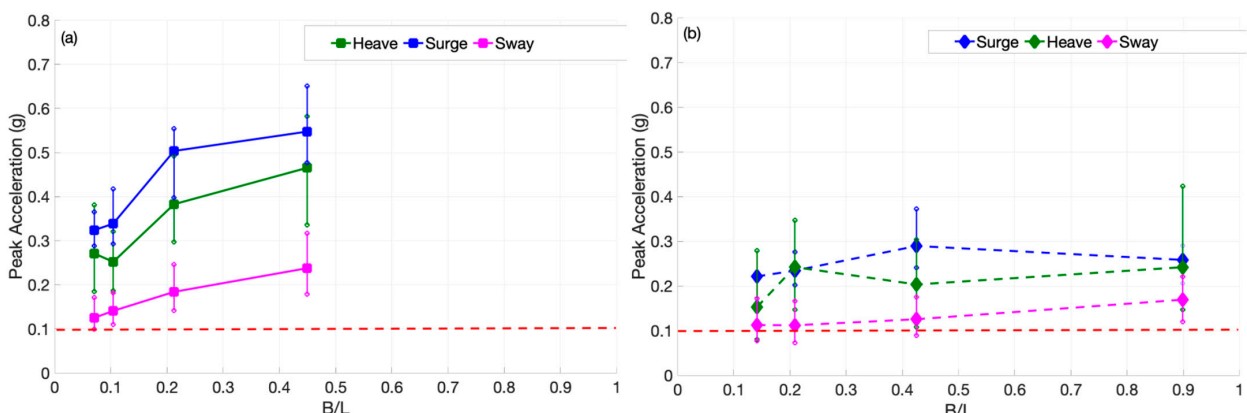

**Figure 7.** Peak in Single (Surge, Heave and Sway) Axis of Acceleration Plotted Against Beam-to-wavelength Ratio and Compared Against the Safe Motion Limit of 0.1g. (**a**) Narrow Pontoon and (**b**) Wide Pontoon. Range between 5 sensors and 3 test repetitions shown by vertical lines and solid symbol being the average of the 15 results.

While peak accelerations shown in Figure 7 exceed the SML adopted for this study, examining the cumulative distribution functions provides further insight into the probability that a person standing on a floating pontoon would experience accelerations that exceed the safe motion limit criteria. In general, less than 5% of the data in surge or heave exceeded the peak SML = 0.1g (Figure 8). This suggests that the peaks in acceleration (Figure 7) resulted from infrequent, short duration impacts due to the pontoon/pile interaction (Figures 5 and 6) rather than the interaction with the incoming wave that was observed to be minimal when examining the reflection and transmission coefficients (Figure 4). When considering the linear vector accelerations of all three axes combined (Figure 8c), the mean probability of exceeding the peak SML = 0.1g was as high as 12% for the 3 s wave on the Wide pontoon (B/L = 0.43).

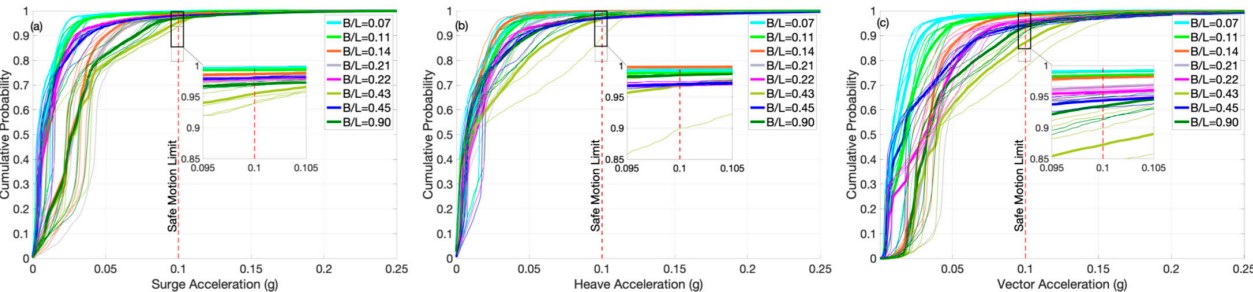

**Figure 8.** Cumulative probability distributions of measured accelerations of all 5 sensors for (**a**) surge; (**b**) heave; and (**c**) vector.

### 4.3. Comfort Criteria: Root Mean Square (RMS) Acceleration

While peak accelerations are important to understand with respect to patron safety on floating pontoon structures, engineers must also consider the overall general movement and a person's comfort (Table 1). The RMS acceleration of the piled-pontoon represents overall variability in motion compared to the short duration peak accelerations reported in Section 4.2.

Table 6 summarizes the mean RMS accelerations calculated for each of the axes (x-, y-, z-) based on the triplicate runs. For both pontoons, the highest RMS acceleration in both surge and heave was recorded when the beam was almost half the wavelength (B/L = 0.43 and 0.45). Similar to the observed peak accelerations (Figure 7), the RMS acceleration for surge exceeded the comfort SML (0.03g) for all tests and was as high as 0.09g. Heave RMS accelerations exceeded the SML (0.02g) for all tests apart from the 7 s period waves for both pontoons (B/L = 0.07 and 0.14). The RMS sway (y-axis) acceleration did not exceed the SML (0.03g) criteria for any of the scenarios tested. These results indicate that accelerations exceeding the SML in the direction of wave propagation (surge) and vertically (heave) for the cases tested here are primarily due to the pile-pontoon interaction rather than the wave itself (Figure 5) and are consistently large enough to cause discomfort for passengers using floating pontoons exposed to relatively small monochromatic boat wake.

**Table 6.** Root Mean Square (RMS) Acceleration in x-, y-. and z-axis for each of the tested wave periods for Narrow and Wide Pontoons. All values given in g. Bold indicates exceedance of SML.

| Axis and Test ID | SML Acceleration Criteria (g) | B/L | | | | | | | |
|---|---|---|---|---|---|---|---|---|---|
| | | 0.07 | 0.11 | 0.14 | 0.21 | 0.22 | 0.43 | 0.45 | 0.90 |
| **Test ID** | | B4 | B3 | B8 | B7 | B2 | B6 | B1 | B5 |
| ax surge | 0.03 | 0.04 | 0.04 | 0.04 | 0.06 | 0.05 | 0.07 | 0.09 | 0.05 |
| ay sway | 0.03 | 0.01 | 0.02 | 0.01 | 0.01 | 0.02 | 0.02 | 0.03 | 0.02 |
| az heave | 0.02 | 0.02 | 0.06 | 0.02 | 0.03 | 0.06 | 0.04 | 0.06 | 0.03 |

*4.4. Frequency of Acceleration*

For public access floating pontoons it is also important to assess the frequency of acceleration as humans are more likely to have an unfavourable response to motions within a frequency band of 1–80 Hz [30–32]. Frequency domain filtering of the raw acceleration time series of the pontoons indicate that as the cut off frequency is increased the maximum observed peak acceleration increases. For the Narrow pontoon, accelerations exceeded the SML (0.1g) with frequencies above 1 Hz for all wave periods (2 to 7 s). For the Wide pontoon, the 3 s period wave exceeded the SML at a frequency of 1 Hz, while the other wave periods tested did not exceed the SML until a frequency of approximately 2 Hz. Additionally, [31,45] identified that there was a peak in postural instability at approximately 0.5 Hz. Maximum peak acceleration at a frequency of 0.5 Hz also exceeded the SML in the Narrow pontoon for B/L = 0.22 (0.12g in surge and 0.11g in heave) and for the Wide pontoon for B/L = 0.43 (0.10g in surge and 0.11g in heave).

*4.5. Angles of Motions*

Angles of rotation about the horizontal axes of floating pontoons are also an important design aspect that should be considered. Both the peak angle limit (operational SML = 6°), which may induce tipping, and the RMS angle limit (comfort criteria SML = 2° RMS), which refers to overall variability are considered here. Given the unidirectionality of the wave in the 2D flume and the piles constraining angular motion, roll (about the y-axis) was the primary angle of motion of the pontoons. Roll may be affected by a combination of natural period of roll (Table 4), wave steepness (H/L), beam-to-wavelength (B/L), beam-to-draft (B/D) (Table 2) and pontoon-wave interaction.

Analysis of the results identified roll rotations above the recommended operational SML 6° (Peak) and comfort SML 2° (RMS) limit (Figure 9a,b). For the Narrow pontoon (Figure 9a) both peak and RMS SML criteria were only exceeded after B/L exceeded 0.2 (wave period less than 5 s). For the Wide pontoon (Figure 9b) both peak and RMS SML criteria were exceeded for all tests other than the 2 s wave period (B/L = 0.9). For both pontoons, the highest roll angles observed corresponded to when the pontoon was observed to hang on the piles as the crest of the wave pushed up the front face of the pontoon (e.g., Figure 6). Comparing these results to the natural periods in heave and roll (Table 4) for both the Narrow and Wide pontoons, the roll response is not expected to have been excited adversely by the incident waves. Instead, both pontoons had the largest observed roll when B/L ~0.45.

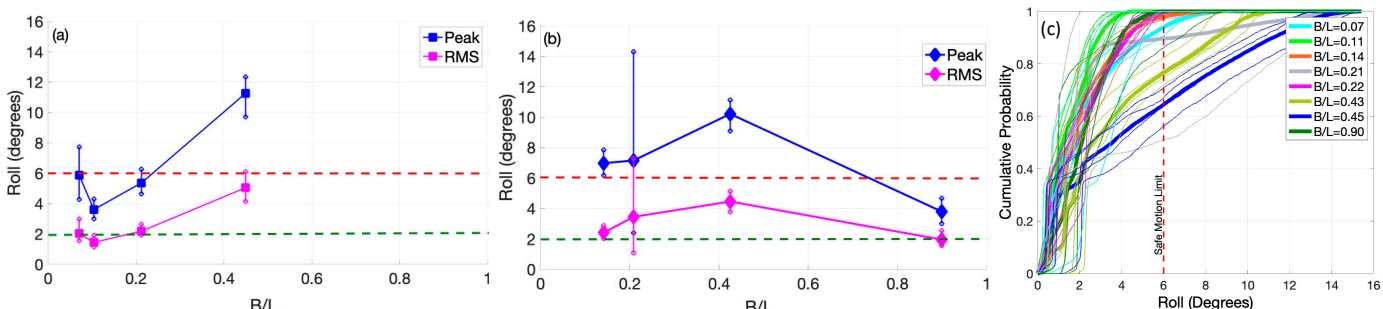

**Figure 9.** Peak and RMS Roll Plotted Against Beam-to-wavelength for: (**a**) Narrow Pontoon and (**b**) Wide Pontoon. Upper red dashed lines show peak SML and lower green dashed line RMS SML. Range between 5 sensors and 3 test repetitions shown by vertical lines and solid symbol being the average of the 15 results. (**c**) Presents the CDF of the Peak Roll for all sensors and tests.

Examining the cumulative distribution functions of roll angles (Figure 9c) there is a clear dependency in roll exceedance on B/L, with B/L approaching half a wavelength resulting in significantly higher exceedance of the SML = 6 degrees (36.27%, B/L = 0.45 Narrow and 23.22%, B/L = 0.43 Wide pontoon).

## 5. Discussion

### 5.1. Comparison to Field Measurements

The focus of these experiments was on the dynamic response of a floating piled pontoon under idealized boat wake conditions. While a Froude similitude with a length scale of 10 was chosen, scale effects should be considered. Freeman et al. [46] present preliminary findings from field testing of four pontoons around Sydney and the Shoalhaven, NSW, Australia. Orient Point pontoon, located in the Shoalhaven, most closely resembles the scaled laboratory results presented here for incident wave conditions (H = 0.3 m and T = 2 s). Orient Point is larger than the laboratory pontoons with a displacement of 18 tonnes, Beam-to-draft (B/D) ratio of 7.2, and Beam-to-wavelength (B/L) = 0.33. Dimensions of the pontoon-pile connection also differ. For comparison, laboratory scale tests B1 and B2 (Table 3) are the most similar to the field conditions of Orient Point. As with the laboratory tests presented above, field tests also exceeded the prescribed SML. Overall magnitudes of peak accelerations agree between field (~0.2–0.5g) and lab (~0.2–0.55g), however the field data showed maximum accelerations in the lateral direction rather than heave. RMS accelerations were also comparable between field (0.01–0.09g) and lab (0.02–0.09g), with both showing maximum in the surge axis. In general, the agreement between field and lab data suggests the laboratory results presented here are reasonable in terms of expected magnitudes of accelerations that would be observed in the field. Several factors may contribute to the differences in the axes of maximum acceleration including pontoon draft, dimensions, and pontoon-pile connection. Both lab and field results indicated that the peaks in lateral acceleration were resultant from the pontoon being pushed against the pile, while peaks in the vertical accelerations in the lab were linked to the pontoon hanging after the wave passes and then falling. The pontoon-pile connection—specifically the gap space between the pontoon collar and pile was not a focus of either the lab or field testing but these results suggest they should be considered in future work.

### 5.2. Safe Motion Limit Criteria

Floating pontoons as shown in Figure 1 are public access structures and as such, the comfort and stability of patrons should be considered during the design phase. Preliminary patron surveys undertaken during the field testing of pontoons in Sydney Harbour indicate users experience both motion sickness and discomfort [46]. Notably, field conditions presented in [46] were for predominately milder (less steep) waves and much larger pontoons than those presented here, yet patrons still experienced motions they deemed uncomfortable. In field-based situations waves can be multi-directional and a result of multiple coinciding boat wakes, as well as wind-generated waves producing far more complex seas and resultant dynamic motions. Therefore, the laboratory results of 2D dynamic motions presented here resulting from monochromatic, uni-directional waves are idealized, with patrons likely to be more adversely affected in field-based situations by complex wave environments. Additionally, the safe motion limits adopted for this study were based on literature describing able-bodied adults. Young children (<7 years) and the elderly (>65 years) also frequent public wharves and have significantly lower stability limits [24]. Considering that floating pontoons are public access structures, we advocate that the safe motion limit criteria presented here should be considered as a guideline for upper limits in design.

### 5.3. Beam-to-Wavelength Ratios (B/L)

Beam (B) to wavelength (L) is an important parameter that is considered by coastal engineers when designing marine structures such as floating pontoons. For small B/L, the structure will ride on the incident wave, resulting in accelerations related to the incoming wave, very little reflection, and nearly 100% transmission (Figure 4). Gaythwaite [40] identified that at a beam-to-wavelength ratio of 0.2 or less, a floating breakwater essentially follows the wave contour with little or no wave attenuation. This agrees with the laboratory

data presented here, where lower values of B/L between 0.07 and 0.22 (wave periods between 5 and 7 s) saw nearly 100% transmission (Figure 4b). In contrast, large B/L (shorter waves) resulted in high reflection, low transmission (Figure 4a,b) and accelerations related to the interaction between the structure and the wave (Figure 7). Results indicate the most adverse motion response was observed when the beam approached half the wavelength (B/L = 0.43 and 0.45). The results presented here suggest that for optimizing patron comfort and safety, B/L is an important design consideration and ratios approaching 0.5 should be avoided to limit adverse accelerations and roll angles.

## 6. Conclusions

Floating pontoons are commonly used as public access structures in small craft harbours and as such, the comfort and safety of patrons must be considered during the design phase by coastal or maritime engineers. Here, a new set of physical laboratory experiments were presented that specifically examined the dynamic motions of two different piled box-type floating pontoons of varying beam width under monochromatic boat wake conditions with periods ranging from 2 to 7 s. The dynamic motions (accelerations and roll angles) were compared to safe motion limit criteria as defined in the literature for personal safety and comfort. The most energetic behaviour occurred for beam-to-wavelength (B/L) ratios between 0.4 and 0.5, where there was visible wave-pontoon and pontoon-pile interaction. Notably, the most adverse conditions recorded in acceleration and roll were due to pile-pontoon interaction as the pontoon was pushed against the piles or 'hung' off the piles as each wave passed. These consistent, but short-lived high accelerations resulted in peak accelerations in heave and surge more than six times the peak acceleration SML (0.1g) and up to 6 times the limit in RMS accelerations. Roll rotation above the 6° SML was also observed for both pontoons. Encouragingly, despite the high peaks in acceleration observed, both pontoons had only a 5% occurrence of exceeding the nominated peak safe motion limit SML of 0.1g in heave and surge, indicating that these spikes in acceleration were short-lived.

Laboratory results compared well with preliminary field testing at a single sheltered small craft pontoon exposed to boat wake with respect to the peak and RMS accelerations observed. In more diverse field situations, where multiple boat wakes may interact with each other forming complex 3D seas, as well as the presence of wind generated waves, pontoon motions are expected to be more complex. The results presented here highlight the need for more detailed understanding of the dynamic motions of public access structures, such as piled floating pontoons in order to fully consider public comfort and safety.

**Author Contributions:** Conceptualization, E.L.F., K.D.S., R.J.C.; methodology, E.L.F., K.D.S., R.J.C.; formal analysis, E.L.F.; writing—original draft preparation, E.L.F., K.D.S.; writing—review and editing, K.D.S., R.J.C., F.F.; visualization, E.L.F.; supervision, K.D.S., R.J.C., F.F. All authors have read and agreed to the published version of the manuscript.

**Funding:** This research received no external funding.

**Institutional Review Board Statement:** Patron surveys were collected under UNSW ETHICS HC20003.

**Informed Consent Statement:** Informed consent was obtained from all subjects involved in the study.

**Data Availability Statement:** Data may be made available upon request to the corresponding author.

**Acknowledgments:** This research was completed in fulfillment of a MPhil degree by E.F. and she was funded by the Australian Research Training Program. The work present here is included within her full thesis. A portion of this work has also been included in the 2022 First Place Paper in the PIANC De Paepe-Willems Award for Young Professionals submitted by E.F.

**Conflicts of Interest:** The authors declare no conflict of interest.

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
