# Peer review of "Dynamic Motions of Piled Floating Pontoons Due to Boat Wake and Their Impact on Postural Stability and Safety"

_jmse, doi:10.3390/jmse10111633_

Round 1

Reviewer 1 Report

The manuscript is interesting and its topic is a good fit for JMSE. It is well-written and technically sound. I have only a few comments as shown below, which help improve the manuscript further:

·        Figure 2:  Can you please draw on top of these figures the axis x,y, z and the 6motions (surge...roll etc)? Such a visualization would make the manuscript much more comprehensible.

·    Line 214: How did you select the recording rate of 50Hz? How are you sure that this frequency is adequate to capture the impulsive acceleration peaks?

·     Figure 4. It is surprising to see that the width of the structure has nearly no effect on the kt and kr. Why is that? I am wondering how the total forces on the two width cases look like. On one side it could be assumed that the structure with larger width would witness larger hydrodynamic (uplift) loads. On the other hand, a recent study (Xiang et al, 2021) showed that after a certain value of Lwave/L the forces do not increase anymore. So, how do the current findings (of Figure 4) and of section 5.3 relate to the findings of the aforementioned study?

Xiang et al (2021): Assessment of Extreme Wave Impact on Coastal Decks with Different Geometries via the Arbitrary Lagrangian-Eulerian Method. Journal of Marine Science and Engineering, 2021, MDPI, 9(12), 1342;

·   Lines 352-354: The authors mention here that the pontoon is pushed against the piles generating impact accelerations. Wouldn't it be interesting to look at the loads on the piles? Since the pontoon exceeds the accelerations of 'comfort' prescribed by the available codes, couldn't the impact loads on the piles also exceed the design recommendations?

·        Line 432-433: The authors mention here the unidirectionality of the wave and the 2D nature of the flume, which implies that they did not test any oblique wave-pontoon interactions. It is traditionally thought that oblique cases results in less severe loading/response conditions, which explains why 95% of the studies available in the literature focus on the normal impact of a wave on a structure. However, recent work (Istrati and Buckle, 2021) revealed that the 3D effects generated in cases where the waves and the main axis of the platform/deck are not aligned, have a complex effect because they can reduce the hydrodynamic forces in one direction but generate additional out-of-plane forces, yaw and roll moments, which should be taken into consideration in the design of any structure in the marine environment. Given the significant 3D effects demonstrated in the aforementioned studies, and the fact that the direction of the waves in a port can change, the reviewer would advise the authors to discuss briefly the importance of the relative angle as seen in the literature and point out that the present study focused only on one wave direction, and it is currently unknown if there are other ‘worse’ scenarios (that might occur when the angle is different)

Istrati, D., Buckle, I.G. (2021): Tsunami Loads on Straight and Skewed Bridges–Part 2: Numerical Investigation and Design Recommendations (No. FHWA-OR-RD-21-13). Oregon. Dept. of Transportation. Research Section, https://rosap.ntl.bts.gov/view/dot/55947

·        Line 478-484: The authors mention here the importance of the pontoon-pile connection and the need to investigate it in the future. The reviewer agrees totally with this suggestion, because after the pontoon pushes/’hugs’ on the pile then the connection will be very stiff and certain connections/piles might be overloaded (hydrodynamic load not uniformly distributed to the two piles), which would result in higher probability of failure as was observed in other relevant applications, such as, in the case of wave impact on coastal decks (Istrati et al, 2018).

Istrati et al (2018): “Deciphering the tsunami wave impact and associated connection forces in open-girder coastal bridges”, Journal of Marine Science and Engineering, 2018, MDPI, 6 (148)

Reviewer 2 Report

Review of the manuscript by Freeman et al.

Dynamic Motions of Piled Floating Pontoons due to Boat Wake and Their Impact on Postural Stability and Safety

Submitted to Journal of Marine Science and Engineering (1912293)

GENERAL COMMENTS

The paper deals with the vibration behaviors of the piled floating pontoons. A set of laboratory-scale physical model experiments of two varying beam width piled floating pontoons subjected to boat wake conditions have been presented. Some parametric studies have been given, and some complex interactions have been revealed. Moreover, some laboratory results have also shown and compared, and a good agreement has been observed. According to the high-quality standards of the Journal of Marine Science and Engineering, the paper can be considered for publication after some revisions. Some major and minor comments are summarized as follows.

MAJOR/MINOR COMMENTS

u  The INTRODUCTION is essentially a disordered list of concise (sometimes vague and imprecise) statements about what other authors did in the past in the field. The mere sum of these statements is far from a coherent analysis of the current state of the art and certainly does not suffice to support the paper’s motivations. The authors must relate the referenced works to each other by highlighting the advancements and significant theoretical/applied results of each piece of research. The motivation of the manuscript should be illustrated more clearly and concisely.

u  The readability of these figures and tables should be improved. Many parametric studies are presented, and I am wonder why these types of parameters are selected and studied.

u  The experimental procedures and results should be illustrated more specifically.

u  The waves seem easy to be broken, so I wonder how the authors consider these breaking waves’ effects.

u  Some parameters and coefficients should be explained more clearly.

u  If more wave periods are considered here, I wonder what the results’ differences are.

u  It seems that there is nothing new in some sections, so some reductions should be made.

u  The innovation and creativity should be illustrated more clearly in the Conclusion. It seems that some conclusions are not new.

Reviewer 3 Report

Paper No. JMSE-1912293-v1

Title: Dynamic Motions of Piled Floating Pontoons due to Boat Wake and Their Impact on Postural Stability and Safety

The study is very useful for design of floating structures engineering and or very large floating structures, considering general movement to a person's comfort. I would therefore recommend publication, after some changes are made.

1. What method is adopted in reflection coefficient estimation? e.g. Goda, one-point, two-point, three-point...., please include more detail.

2. Reflecting wave should affect wave motion obviously. In this research, what facility is installed in the end of flume to reduce wave reflection. The reflecting wave should be removed or the effects of reflecting wave should be analyzed.

3. It is recommended not to use the period T for the x-axis in Figure 4, and parameters related to the wave length should be used, such as parameters such as B/L, H/gT2.... in Figure 7, the interaction between the wavelength and the structure itself is clear, however, the "corresponding wavelengths" of the period of the equidistant difference are not equidistant, the relationship between the structure and the wavelength should be more accurately represented.
